# Cell surface crowding is a tunable energetic barrier to cell-cell fusion

Daniel S. W. Lee[1,6], Liya F. Oster[1,2,6], Sungmin Son[1,5] & Daniel A. Fletcher [1,2,3,4] ✉

Cell-cell fusion is fundamental to processes such as muscle formation and viral infection. An essential step in fusion is close membrane apposition, but cell membranes are crowded with proteins, glycoproteins, and glycolipids, which must be cleared before a fusion pore can be nucleated. Here, we find that cell surface crowding reduces fusogenicity independent of how fusion is driven. We estimate that crowding presents an energetic barrier to membrane apposition on the scale of ~$100k_BT$, greater than that of bare membrane fusion alone. We show that increasing cell surface crowding reduces fusion efficiency of PEG-mediated and fusogen-mediated cell-cell fusion, as well as synthetic membranes under force. Interestingly, we find that differentiating myoblasts naturally decrease their surface crowding prior to fusion. In this work, we show that cell surface crowding presents an underappreciated biophysical barrier that may be tuned developmentally and could be targeted externally to control tissue-specific cell-cell fusion.

Cell-cell fusion is critical for the formation of specialized multi-nucleated cells in varied biological systems. In vertebrates ranging from lamprey eels to humans, fusion of myoblasts is required for development and maintenance of skeletal muscle fibers[1]. Similarly, multinucleated osteoclasts are crucial for bone resorption[2]. In mammals, fusion of mononucleated trophoblasts into the multinucleated placental syncytiotrophoblast provides a protective environment for embryo development[3]. Some viruses are also known to cause fusion of host cells. Fusion of respiratory epithelial cells by the SARS-CoV-2 virus can indicate severity of an infection[4], and members of the orthoreovirus family cause fusion in their infected hosts, aiding in viral spread[5,6]. Each of these physiologically diverse examples of cell-cell fusion is driven by distinct membrane fusion proteins, called fusogens. Typically broken into three major classes based on their different structures, fusogens function through a wide variety of mechanisms, including disrupting bilayer packing to facilitate lipid mixing of apposed bilayers, inducing bilayer curvature favorable for hemifusion stalk or pore formation, and recruiting necessary accessory proteins to the fusion synapse[7–10].

Despite structural and mechanistic differences in fusogens, lipid bilayers that fuse are believed to follow a qualitatively similar pathway to overcome the considerable energetic barrier to fusion, which for bare lipid bilayers has been estimated to be ~$40k_BT$[11,12]. First, the membranes must come into close apposition. They then undergo hemifusion, in which the outer membranes fuse, forming a stalk, followed by full fusion, in which the hemifusion stalk opens into a fusion pore[11]. While this pathway is consistent with in vitro membrane fusion experiments and simulations[13,14], it leaves out aspects of real cell-cell interactions that could significantly affect this fusion landscape and resulting fusion efficiency. In particular, the first step, during which membranes are brought into close apposition, requires exclusion of surface molecules and solvent from the intermembrane space (Fig. 1A). This raises the question of whether cell surface crowding is a significant barrier that could limit cell-cell fusion.

The eukaryotic cell surface is decorated with a forest of glycosylated proteins and lipids, known as the glycocalyx[15,16] at densities on the order of 20,000 molecules per square micron[17]. These surface glycoproteins must be locally excluded from the site of fusion,

[1]Department of Bioengineering, University of California, Berkeley, CA, USA. [2]Graduate Group in Biophysics, Berkeley, CA, USA. [3]Division of Biological Systems and Engineering, Lawrence Berkeley National Laboratory, Berkeley, CA, USA. [4]Chan Zuckerberg Biohub, San Francisco, CA, USA. [5]Present address: Department of Bio and Brain Engineering, Korea Advanced Institute of Science and Technology, Daejeon, Republic of Korea. [6]These authors contributed equally: Daniel S. W. Lee, Liya F. Oster. ✉e-mail: fletch@berkeley.edu

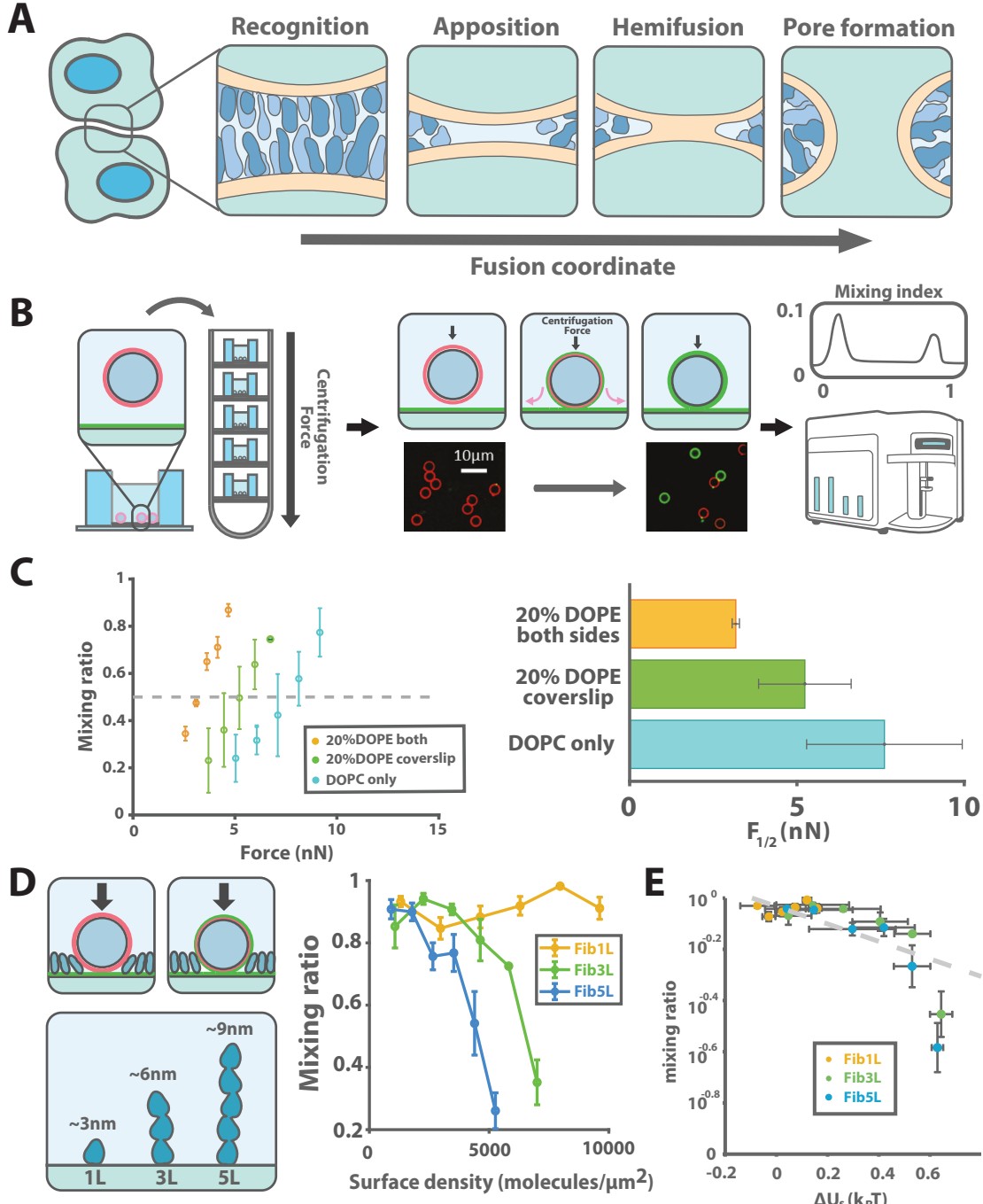

**Fig. 1 | Surface crowding poses a barrier to membrane fusion in vitro. A** Cell-cell fusion requires close membrane apposition and surface protein displacement. **B** Our reconstituted membrane fusion assay applies force via centrifugation to drive membranes (red) on a bead in solution and on a surface (green) together. The mixing ratio, defined as the ratio of the surface to bead SLB fluorescence (Liss Rhodamine PE and Atto488-PE respectively) between the bead and surface bilayers is dependent on the force applied and lipid composition of the respective surfaces. **C** To test the effect of surface protein composition on membrane fusion, the planar SLB was decorated with purified proteins of varying size and fused to beads with a constant force of 5nN. The mixing ratio was calculated for each force condition, and linear interpolation gave an estimate of $F_{1/2}$ (dashed grey line), which was

compared between three lipid conditions and plotted. Error bars represent standard deviation (SD) in mixing ratio over 3 technical replicates and were propagated by linear interpolation from experimental error. **D** The mixing ratio was averaged over 3 technical replicates for surfaces decorated with varying densities of Fibcon repeats containing 1, 3, or 5 domains (Fib1L - yellow, Fib3L - green and Fib5L - blue lines). Error bars reflect SD. **E** Mixing ratio for each protein density and type was compared to the crowding energy for the corresponding condition to collapse the data, which was fit to a linear model (dashed grey line). Horizontal error bars reflect standard error of the mean (SEM) over 3 replicates for crowding measurements, while vertical ones reflect SD of mixing ratio.

potentially presenting a steric barrier that may hinder cell-cell fusion. Consistent with this idea, the inclusion of pegylated lipids has been found to inhibit liposome fusion in vitro[18], and cross-linking cell surface glycoproteins with multivalent lectins can prevent virus fusion and infection[19]. However, because surface glycoproteins range in height from <1 nm to nearly 1000 nm tall[20], surface density alone is a poor metric of the steric barrier that could resist local exclusion of surface proteins, hindering cell-cell fusion[19]. To address this gap, we recently developed a method to quantify the steric barrier presented by the cell surface, which we refer to as crowding, by measuring the binding affinity (or dissociation constant) of an antibody for an exogenous sensor molecule integrated into a lipid membrane[21]. We previously showed that the dissociation constant of an antibody for the sensor on a live cell surface can be 10 times or more greater than it is on a bare surface, corresponding to a $2 - 3k_BT$ barrier against antibody binding created by cell surface molecules[21]. This suggests that cell surface crowding could be a nontrivial and underappreciated barrier to cell-cell fusion, one that could potentially be modulated to regulate fusion rates.

In this work, we combine cell surface crowding measurements with in vitro reconstitution and live-cell fusion assays to demonstrate that cell surface crowding significantly reduces the efficiency of cell-cell fusion. We first develop an in vitro fusion assay to measure the force required to fuse reconstituted membranes and show that adding surface proteins slows fusion in a manner quantitatively consistent with the steric barriers they impose. We then demonstrate that this principle applies in PEG-mediated fusion assays, where cells with less crowding showed greater likelihood of fusion. In the case of cells expressing viral fusogens, we find that differences in cell surface crowding can result in the selective fusion of less crowded cells, spatially segregating crowded (unfused) cells away from multinucleated syncytia of less crowded cells. Finally, we demonstrate that surface crowding is naturally reduced in differentiating myoblasts as they approach a fusogenic state, suggesting that surface crowding can be modulated by cells in physiological contexts to control the timing of cell-cell fusion.

## Results

### A reconstituted membrane mixing assay that fusion depends on membrane surface crowding

We first sought to investigate the relationship between surface crowding and membrane fusion by developing an in vitro assay to quantify lipid mixing, a step toward full membrane fusion, as a function of force. To collect large numbers of events simultaneously, we used centrifugation to exert a compressive force between 3.8-micron glass beads coated with supported lipid bilayers (SLBs) and an SLB-coated glass coverslip (Fig. 1B). The SLBs were composed of DOPC (1,2-dioleoyl-sn-glycero-3-phosphocholine), with the bead SLB labeled with Liss-Rhodamine PE and the coverslip SLB labeled with Atto488-PE, such that lipid mixing between the membranes resulted in the simultaneous loss of Liss-Rho and gain of Atto488 signal around the bead after 20 minutes of centrifugation. The ratio of Liss-Rho to Atto fluorescence was quantified by flow cytometry, giving a generally bimodal population (Supplementary Fig. 1), which was used to calculate the mixing ratio, i.e., the fraction of beads found to have undergone lipid mixing for a given force. We then calculated the $F_{1/2}$, the force required to give a mixing ratio of 50%, where membrane fusion had occurred in 50% of the beads in the assay (Fig. 1C). To validate that this approach captures known effects of specific lipid species on membrane fusion, we compared the DOPC membranes to SLBs containing 20% DOPE (1,2-Dioleoyl-sn-glycero-3-phosphoethanolamine), a lipid that induces negative curvature. We found that including DOPE in the coverslip membrane decreased $F_{1/2}$ from 7.6 ± 2.3 nN to 5.2 ± 1.3 nN, while including DOPE in both the bead and coverslip membranes further decreased $F_{1/2}$ to 3.17 ± 0.1 nN, consistent with previous work showing that DOPE increases lipid mixing[22] (Fig. 1C).

Next, to quantify the effect of surface crowding on membrane fusion, we decorated the coverslip SLB with purified His-tagged proteins formed from repeats of fibronectin type III consensus (Fibcon) domains using Ni-NTA lipids (Fig. 1D), which we had previously used to create synthetic surface proteins with variable known heights[23,24]. We found that the single Fibcon domain protein (Fib1L), which is ~4.2 nm tall, was insufficient to significantly change the mixing ratio, up to the maximum surface density allowable (~10,000 per square micron) by the SLB without disrupting the lipid bilayer, which is below the physiological protein density on the cell surface of about 20,000 per square micron[17]. In contrast, decoration with the ~7 nm tall triple Fibcon domain protein (Fib3L) decreased the mixing ratio 50% at a density of around 6500 molecules/μm², and only about 4500 molecules/μm² of the 10 nm quintuple Fibcon (Fib5L) domain protein was required to reduce the mixing ratio to 50% (Fig. 1D). These results suggest that not only the density but also the size of the surface protein is important for its ability to inhibit fusion.

We hypothesized that the steric barrier presented by cell surface crowding could directly explain the decrease in membrane fusion in vitro. We previously showed that the energy penalty for insertion of a molecule into a polymer brush, here defined as $\Delta U_s$, can be read out by measuring the apparent dissociation constant ($K_d$) of an adsorbed molecule for the surface in question compared to a bare surface ($K_d^0$)[21], where $K_d = K_d^0 \exp(\Delta U_s/k_BT)$. Since the fusion interface must be entirely cleared of both bystander and solvent molecules for fusion to proceed, the fusion rate should simply be proportional to the crowding energy $\Delta U_s$, if crowding is the limiting step to fusion. Using previous measurements of $K_d/K_d^0$ for Fibcon domain repeat proteins on SLB-coated beads (Supplementary Fig. 1), we estimated $\Delta U_s$ and compared it to the mixing ratio for each surface density and protein size condition. This revealed a strong correlation between mixing ratio and crowding energy (Fig. 1E), suggesting that surface crowding quantitatively alters membrane mixing in our reconstituted assay.

### Lowering surface crowding enhances PEG-mediated fusion

Next, we investigated the effect of cell surface crowding on fusion of live cells. We used polyethylene glycol (PEG) to drive cell-cell fusion in the absence of any fusogen, a common practice for generating hybridomas for monoclonal antibody production (Fig. 2A)[25–27]. Adding a high concentration of soluble PEG to the media results in an increase in osmotic pressure between cells, bringing them together and driving fusion. To test whether soluble PEG could drive fusion between cells and SLBs, we dissociated HEK239T cells using 2 mM EDTA in PBS and stained them with CellMask Far Red. After allowing them to settle on an Atto488-PE-containing SLB coated coverslip, we added soluble PEG to the settled cells and analyzed lipid mixing by flow cytometry after 20 minutes as previously described. We found that 20% PEG8K, was sufficient to cause lipid mixing between the HEKs and surface SLB (Fig. 2C). Gently trypsinizing the HEKs to lower surface crowding (Fig. 2B) prior to incubation with the PEG-containing solution further increased the degree of mixing observed, suggesting that digesting surface proteins drove lipid mixing (Fig. 2C).

To test the effect of cell surface crowding on content mixing, which occurs with full fusion rather than only lipid mixing, we developed a PEG-mediated fusion assay for suspension cells. Briefly, we pelleted cells with soluble PEG to induce fusion, mechanically dissociated the pellet with vortexting and pipetting to break up aggregates, and then we ran the cells through a flow cytometer with a brightfield imaging capability (Fig. 2D). Using a population of Jurkat cells with half expressing H2B-GFP and the other half expressing H2B-miRFP, we found that PEG-mediated fusion and content mixing was successful with soluble PEG3.5 K (Fig. 2E).

We next tested whether changes in cell surface crowding would alter cell-cell fusion in the PEG-mediated fusion assay. We chose to test K562 cells, which have naturally high cell surface crowding (about

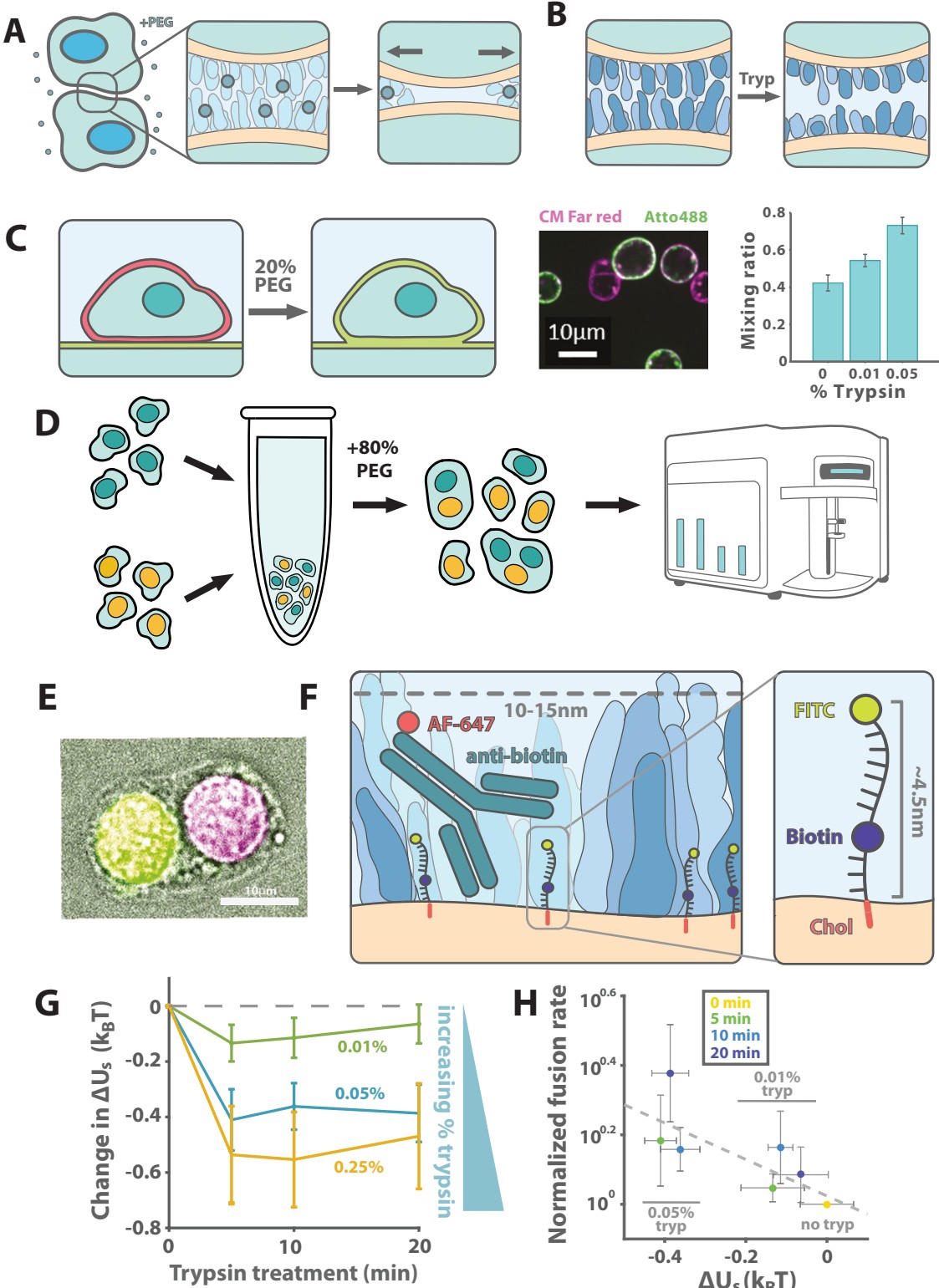

3$k_\text{B}T$, Supplementary Fig. 3) and do not adhere to one another in solution. Furthermore, they are large compared to Jurkat cells, allowing fused cells to easily be identified based on their size using brightfield images taken by the flow cytometer combined with an automated image segmentation pipeline. To reduce crowding by cleaving surface proteins, we incubated cells in dilute trypsin, subsequently quenching with FBS-containing growth media. We found that the decrease in $\Delta U_s$ saturates nearly immediately at a level dependent on trypsin concentration (Fig. 2G). We then tested the effect of these treatments on cell-cell fusion with 0.01% and 0.05% trypsin conditions, reasoning that they would span the range of $\Delta U_s$ values seen in Fig. 2G. Combining the fusion efficiency and surface energy data, we find that the log of the fusion probability decayed with the crowding as expected for changes in surface accessibility alone (Fig. 2H; Supplementary Information), suggesting that crowding may be a limiting factor for cell-cell fusion in some contexts.

**Fig. 2 | Protein crowding is a barrier to PEG-mediated cell-cell fusion. A** PEG was used to exert an osmotic force to fuse cells. **B** Trypsinization can be used to non-specifically lower surface crowding. **C** Centrifugation in the presence of PEG was used to fuse HEK cells (labeled with CM Far Red) to planar SLBs (labeled with Atto488). Cells dissociated with trypsin showed more lipid mixing than those without. Mixing ratio was averaged over 3 biologically independent replicates; error bars reflect SD over 3 biologically independent replicates. **D** Cells were fused by incubating suspension cell lines in 80% PEG solution and analyzed by microscopy or flow cytometry. **E** Jurkat cells expressing H2B-miRFP670 and H2B-GFP were used to validate this approach. **F** Cell surface crowding can be quantified by incorporating a probe consisting of a DNA oligo decorated with cholesterol, biotin, and fluorescein into cell membranes and measuring binding of an anti-biotin antibody. **G** Gentle trypsinization of K562 cells was used to vary surface crowding energy, which was measured with the crowding sensor in flow cytometry and normalized to nontreated control (dashed gray line) after averaging over replicates ($N = 4$ for the 0.01% trypsin condition, 3 for the 0.05% trypsin condition and 1 for the 0.25% trypsin condition). Error bars reflect SEM, propagated over replicate variability and fit error. **H** Fusion rates compared to the crowding energy changes due to a range of trypsinization conditions in K562 hybridomas and fit to a linear model (dashed line; $p = 0.039$ by one-sided $F$ test against a constant model). Horizontal error bars reflect fit SEM for surface crowding measurement; vertical error bars reflect SEM for 3 biologically and technically independent replicates of PEG-mediated fusion assay.

## Surface crowding inhibits FAST-protein mediated fusion

Having determined that cell surface crowding is a barrier to PEG-mediated fusion, we wondered whether it is also a barrier to fusogen-mediated cell-cell fusion. We utilized the fusion associated small transmembrane (FAST) protein p14, which, despite its 1.5 nm height[28,29], is able to drive fusion by recruiting the actin cytoskeleton[30,31]. We previously showed that overexpression of tall proteins, such as the Signal-Regulatory Protein Alpha (SIRPα) ectodomain (~14 nm tall[24]), can increase surface crowding, while overexpression of small proteins, such as Fib1L (~3 nm tall), has little to no effect[21] (Fig. 3A). Thus, to test the effect of crowding on p14-mediated fusion, we co-transfected either the tall SIRPα ectodomain ("ecSIRPα", which lacks the SIRPα cytoplasmic domain) or the short Fib1L, which act as 'bystanders' since they are not required for fusion, with p14 and quantified cell-cell fusion by fluorescence microscopy (Fig. 3B). Strikingly, we found that the tall bystander nearly completely inhibited fusion, while the short bystander had little effect compared to an mCherry transfection control (Fig. 3C).

Based on this result, we wondered whether cell-cell fusion within a population of cells would be biased towards less crowded cells, potentially conferring a degree of specificity and spatial control over the fusion process based only on surface energy barriers. To test this, we designed a synthetic bystander protein with a nuclear reporter, consisting of a SPOT-tagged bystander ectodomain (i.e., either Fib1L or ecSIRPα) with a downstream T2A cleavage site and an eGFP-tagged SV40 nuclear localization signal (NLS) (Fig. 3D). We observed the expected surface localization of the SPOT-label and nuclear localization of the GFP signal by microscopy, and we confirmed proportionality between GFP expression level and SPOT-label binding by flow cytometry (Supplementary Fig. 3).

We quantified the bias in fusion by co-culturing HEK cells expressing either Fib1L or ecSIRPα bystander constructs with HEKs transfected with p14. We observed that once fused, the NLS-GFP signal expressed by a single nucleus would quickly distribute itself across the syncytium (Fig. 3E). We then segmented the cells (Supplementary Fig. 4) and quantified the fraction of cells expressing the respective ectodomains included in the syncytia by calculating the average GFP fluorescence of nuclei inside the syncytia. We then normalized by the average GFP level across all nuclei to conservatively account for expression level differences and found this relative fluorescence was ~20% lower for cells expressing the SIRPα ectodomain compared to those expressing the Fib1L ectodomain, suggesting that expression of the taller ectodomain indeed biased p14 fusion toward otherwise identical cells with lower energy barriers, an indication that cell surface crowding can guide fusion selectivity (Fig. 3F).

## Surface crowding is modulated during myoblast development

Finally, we asked whether cellular systems might already tune cell surface crowding to regulate fusion in a physiological context. We considered the process of muscle development, a highly regulated and essential process in vertebrates wherein progenitor cells differentiate and fuse into myotubes (Fig. 4A). C2C12 murine myoblasts, a well-characterized cell culture model for muscle development[32], elongate after serum starvation and fuse into large syncytial fibers after 4-6 days in culture (Fig. 4B). We monitored the changes in cell surface crowding with our crowding sensor throughout this process of differentiation. Following successful differentiation, which was validated by the observation of fused myotubes (Fig S4), we dissociated the myoblasts with EDTA, incubated the cells with the crowding sensor and antibody, and quantified crowding by flow cytometry. Interestingly, we observed a gradual but steady decrease in the level of surface crowding throughout the differentiation process, with a change in the surface crowding energy barrier of $1 k_B T$ between 0 and 6 days of differentiation (Fig. 4C), indicating that the cell surface becomes significantly less crowded as differentiation proceeds.

To confirm that our measurements were not influenced by syncytia formation itself, which may reduce crowding due to surface-volume ratios and other effects, we also measured crowding in a C2C12 knockout line lacking the myoblast fusogen Myomaker ($Mymk^{-/-}$), which is necessary for cell-cell fusion[33]. Consistent with previous work, we observed elongation and morphological changes consistent with differentiation upon serum starvation, but no fusion (Supplementary Fig. 4). Repeating our crowding sensor measurement on the knockout line revealed a similarly significant decrease in crowding between days 0 and 6 (Fig. 4D). This suggests that the observed crowding changes are independent of syncytia formation and are instead modulated by the differentiation process itself.

## Discussion

In this work, we quantitatively demonstrate that surface crowding hinders cell-cell fusion by presenting a physical barrier to close membrane apposition.

We observed that lipid mixing ratio and cell-cell fusion rate were both strongly correlated with surface crowding energy, suggesting that surface crowding dominates the energetics of fusion both in vitro and in vivo. This data suggests that in physiological membrane fusion, exclusion of surface proteins presents a potentially rate-limiting step in the fusion pathway.

By directly quantifying cell surface crowding, we can estimate the energetic penalty of excluding cell surface proteins during cell-cell fusion. From our measurements showing a penalty of $\Delta U_s \approx 2 - 3 k_B T$ for incorporation of an IgG antibody, with dimensions of ~10-15 nm[34] (Fig. 4E), we estimate the energetic cost of clearing a 100 nm diameter fusion interface to be $\sim 200 k_B T$ (Supplementary Information; Fig. 4F), substantially exceeding the $40 k_B T$ barrier estimated for bare membrane fusion. However, we note here that, while we performed our crowding measurements at 4 °C to avoid probe internalization, fusion physiologically occurs at 37 °C; active and other temperature-dependent processes may affect protein and membrane mobility and organization.

The presence of a robust cell surface barrier preventing neighboring cells from fusing is rather reassuring; spurious fusion of cells in close contact could lead to apoptosis or other anomalous behavior that must be avoided in multicellular organisms. By controlling the

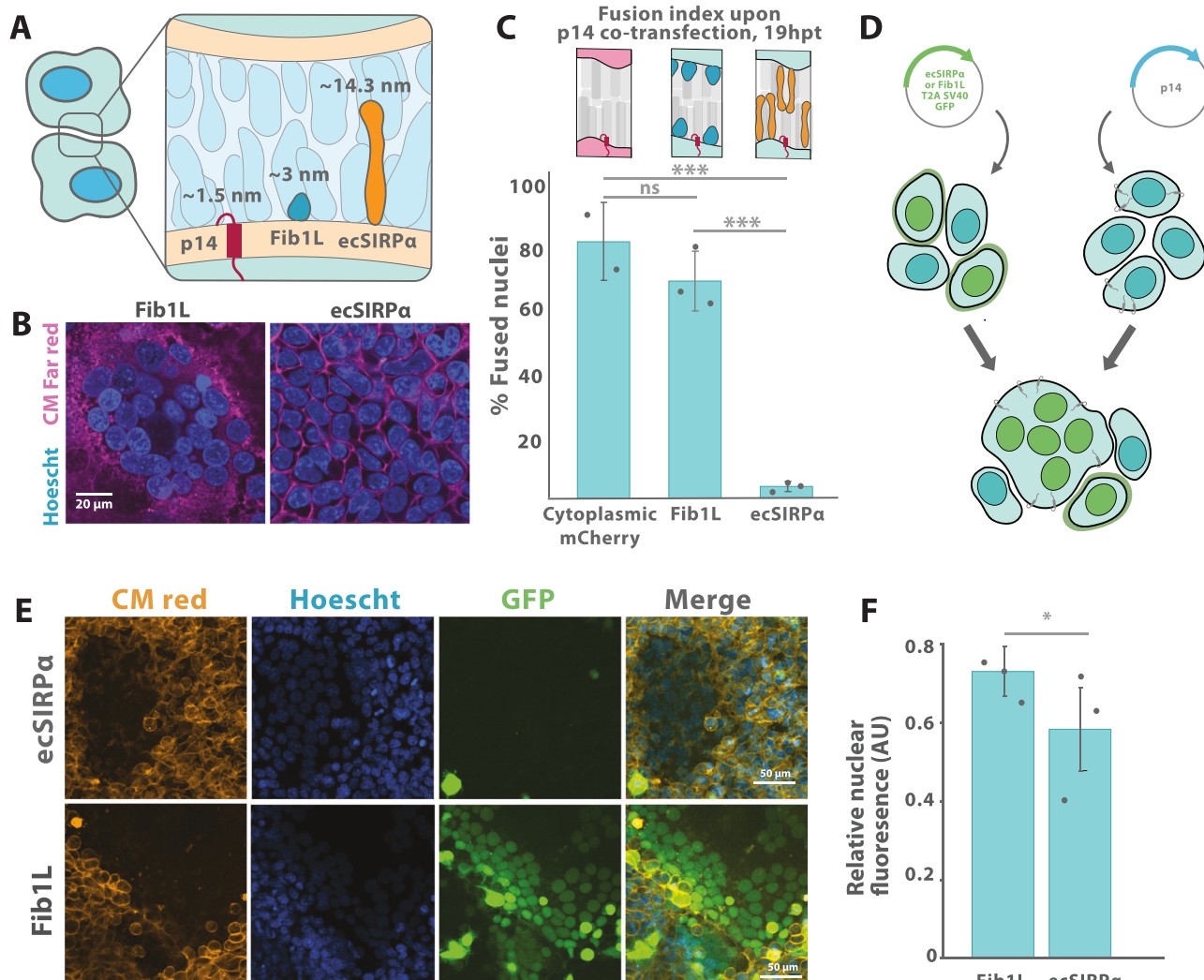

**Fig. 3 | Increased surface crowding inhibits p14-mediated cell-cell fusion, enabling fusion selectivity. A** The FAST protein p14 has been estimated to be 1.5 nm in height, considerably shorter than the bystanders used here. **B** HEK293 cells were co-transfected with p14 and Fibcon1L, SIRPα ectodomain, or cytoplasmic mCherry control. **C** For each condition, the fraction of all nuclei found in syncytia was calculated and averaged over three biologically and technically independent replicates. ecSIRPα was significantly less fusogenic than both F1L and mCherry control ($p = 0.004$ and $0.001$, respectively, by two-sample $t$ test with Bonferroni correction), while F1L was not significantly different from mCherry control ($p = 0.849$). Error bars reflect SEM. **D** The rate of fusion of individual transfected cells was assessed by designing nuclear reporter constructs consisting of an ectodomain and a polycistronic fluorescent protein with an NLS. **E** The NLS-GFP spread between all nuclei within a particular syncytium following fusion for both constructs. **F** The mean nuclear fluorescence was calculated within syncytia, divided by the average nuclear fluorescence per image to normalize for transfection efficiency, and compared between constructs. The propensity of cells expressing ecSIRPα to be fused into syncytia was less than that of cells expressing Fib1L ($p = 0.0214$ by two-tailed $t$ test). Error bars reflect SEM over three biologically independent replicates.

considerable energetic barriers presented by surface crowding, cells can dramatically lower the possibility of cell-cell fusion. Because crowding of the membrane is an emergent property, we anticipate that the biological mechanisms that underlie regulation may be varied, potentially involving hundreds of proteins and diverse pathways, including not only surface glycoprotein expression levels but also other facets of the cellular organization such as the arrangement of the actin cytoskeleton, the metabolic state of the cell, as well as membrane tension, fluidity, and curvature. However, quantifying crowding based upon membrane accessibility allows us to coarse-grain these parameters, whose interplay with crowding should be an avenue for further investigation.

Considering the large barrier cell surface crowding presents to fusion, small changes in crowding could result in large changes in rates of cell-cell fusion. This idea is empirically supported by the observation that gentle trypsinization results in a nearly doubled rate of PEG-mediated fusion, and the $1k_BT$ increase in crowding energy due to ecSIRPα overexpression[21] nearly abrogated p14-driven fusion.

Likewise, crowding is reduced by $-1k_BT$ when comparing pre- and post-differentiation C2C12 myoblasts. In either of these instances, the energetic barrier to fusion is decreased by as much as 50%. Taken together, we speculate that the large energetic barrier posed by cell surface crowding, which can be modified enzymatically or by turnover of the cell membrane, makes this biophysical barrier a biologically convenient mechanism by which cells can modulate fusion.

In summary, we have shown through in vitro and in vivo examples, with and without fusogens, that cell surface crowding is an important constraint on the process of cell-cell fusion, one that could be harnessed for cell engineering and delivery.

## Methods

### Bead preparation and centrifugation
**Bead and coverslip cleaning.** 5um silica microspheres (Bangs Laboratories, Inc, SS05003) were washed in a glass test tube with piranha mixture by adding a 2:3 ratio of 30% hydrogen peroxide

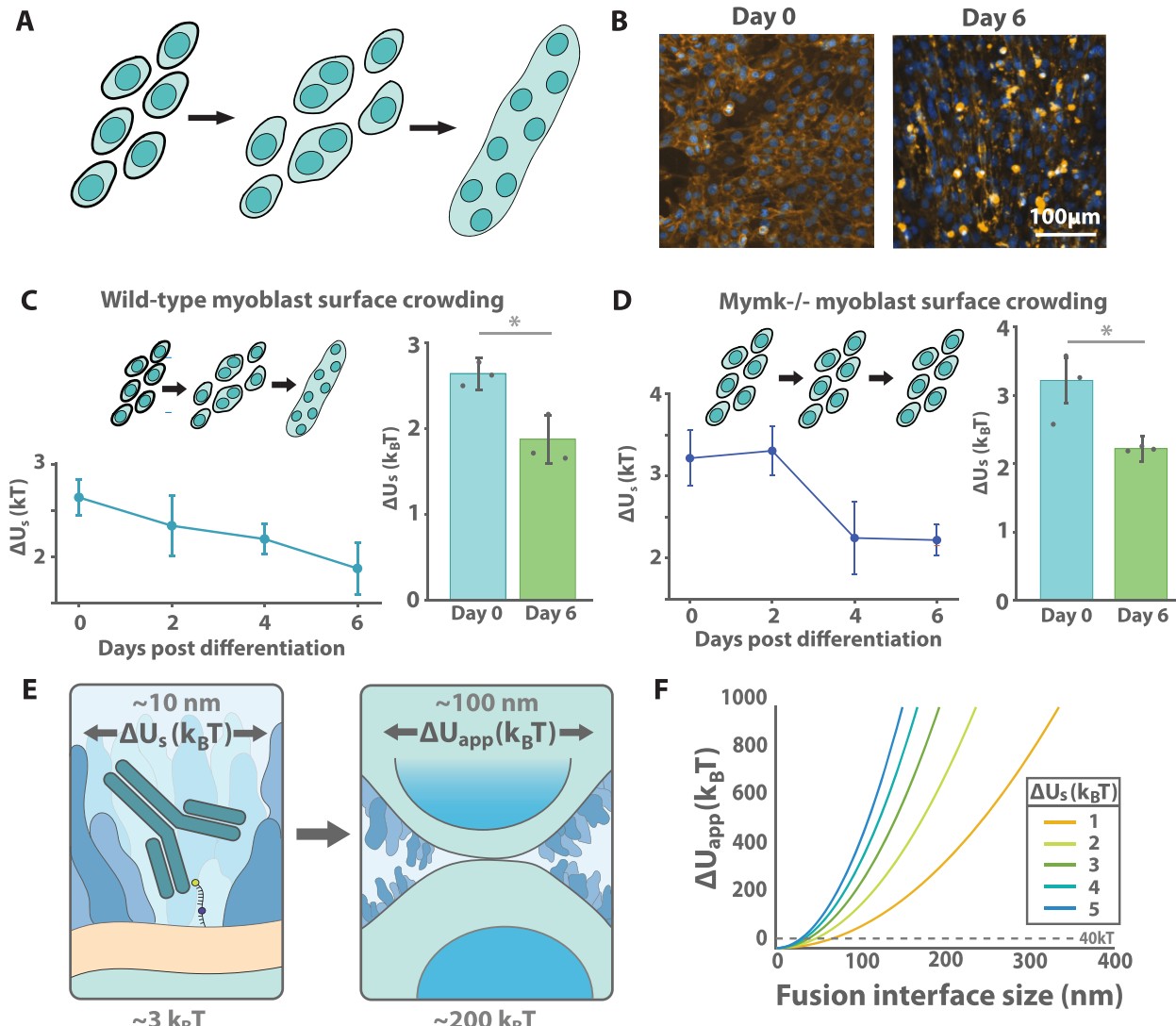

**Fig. 4 | Myoblasts reduce their cell surface crowding as they differentiate and undergo fusion. A** The C2C12 line is a model for myoblast differentiation, during which cells elongate and then fuse. **B** Following serum starvation, wild type C2C12 cells were stained with CellMask Orange and Hoechst 33342, demonstrating myotube formation on day 6. **C** Following serum starvation, crowding was measured in wild C2C12 cells using the aforementioned crowding probe at two-day intervals. Error bars reflect SEM over three biologically independent replicates (*left*). Days 0 and 6 were directly compared and found to be significantly different by two tailed *t* test ($p = 0.012$). **D** Crowding was measured in $Mymk^{-/-}$ C2C12 cells following serum starvation. Error bars reflect SEM over three biologically independent replicates. Days 0 and 6 were directly compared and found to be significantly different by two tailed *t* test ($p = 0.036$). **E** The crowding sensor measures the energy to clear an IgG-sized patch of membrane. We extrapolate this measurement based on parameters from the literature to estimate $\sim 200 k_B T$ for membrane apposition for a 100 nm fusion pore. **F** Full extrapolation of crowding sensor measurement to the energetic cost of membrane apposition ($\Delta U_{app}$, see Supplementary Information).

to full-strength sulfuric acid. The bead and piranha mixture was then sonicated in a bath sonicator for 30 minutes and washed by centrifugation ($1000\,g$) and resuspended in ultrapure MilliQ water (EMD Millipore).

18×18-1.5 mm glass coverslips were mounted onto a teflon rack and RCA cleaned by sequentially boiling them at 80 °C with a mixture of 5 g KOH, 80 mL MilliQ water, and 20 mL of 30% hydrogen peroxide for 30 minutes in a glass beaker. The teflon rack was then briefly rinsed in a beaker of MilliQ water and transferred to another glass beaker of 25 mL full-strength HCl, 80 mL MilliQ water, and 20 mL 30% hydrogen peroxide and again boiled at 80 °C for 30 minutes. Following the acid wash, the teflon rack was transferred sequentially into two beakers each filled with MilliQ water and stored in a third sealed beaker of MilliQ water for later use.

**Supported lipid bilayer formation.** To prepare the lipids, small unilamellar vesicles (SUVs) were formed by mixing the desired lipid composition (POPC, DOPE from Avanti Polar Lipids) in molar ratios in a round-bottom glass tube (0.28 μmol lipid total), and then simultaneously vortexing and desiccating with nitrogen glass to form a thin film at the bottom of the glass tube. To bind the surface proteins, NiNTA-DGS lipid (Avanti) was included in this initial mixture at a molar concentration of 0.1%-7%. SUVs intended for supported lipid bilayer (SLB) formation included 0.01% Atto-488PE, and SUVs intended to coat the glass beads included 0.01% Liss-Rhodamine dye. The glass tubes were then incubated in a vacuum-sealed chamber overnight to ensure evaporation of the chloroform. The dried lipid film was then reconstituted in 900 μL of deionized water, then sonicated with a tip sonicator for 3 minutes on ice with a 2 second duty cycle, amplitude 20%. These SUVs were then stored at 4 °C in 1x MOPS buffer (adding 100 μL of 10X MOPS to the sonicated solution).

To form the bilayers, the SUVs were incubated for 15 minutes in a PDMS well situated on a clean glass slide, allowing them to crash out

onto the glass and form a bilayer. Beads were prepared as described above. Briefly, 40 μL of beads were incubated with 20 μL of silica microspheres in 160 μL MOPS buffer on a rotisserie at room temperature for 20 minutes. Afterwards, the beads were washed 3x via centrifugation at 50 $g$ and resuspended in MOPS buffer. The SLBs were then washed 3x with HEPES buffer. To add proteins to the surface, SLBs containing NiNTA-DGS lipids were incubated with purified His-tagged Fibcon extracellular domain for 15 minutes at room temperature, then washed 3x with MOPS buffer.

**Lipid mixing by centrifugation.** Custom centrifuge inserts were engineered and 3D printed to hold the glass coverslips horizontally in a Type 70 Ti Fixed-Angle centrifuge rotor. 10 μL of lipid-coated beads were added to each SLB well, and the glass slides were mounted on the custom centrifuge inserts. The centrifuge was then spun at a constant speed for 20 minutes. The beads were vigorously pipetted to resuspend all beads, including ones that were fused to the SLB. Imaging the SLB after the bead removal confirmed that less than 1% of the beads remained on the SLB. The supernatant was then run on the Attune CytPix flow cytometer using Attune Cytometric Software. Mixing ratio was then calculated by the fraction of beads that exhibited a gain of green signal, indicating that the green SLB lipids mixed with the lipids coating the bead surface.

## Cell culture and maintenance

All cell lines were acquired from the Barker Hall Cell Culture Facility at UC Berkeley except for C2C12 knockouts, which were a kind gift from Pengpeng Bi (University of Georgia). HEK and C2C12 cells were cultured in T25 cell-culture treated flasks in DMEM media supplemented with GlutaMax, 10% FBS, and 1% pen-strep. Jurkat and HL60 cells were cultured in RPMI with 10% FBS and 1% pen-strep. K562 and Raji B cells were cultured in RPMI with 10% FBS and 1% pen-strep and 1% sodium pyruvate. Cells were split every two days at 1:5 and kept in an incubator at 37 °C and 5% $CO_2$.

C2C12 cells were differentiated by removing growth media, washing with PBS, and replacing differentiation media (DMEM containing 2% horse serum and 1% pen-strep). Cells were gently washed, and media was replaced every 24 h following initial differentiation.

## Plasmid generation

For all plasmids, gene blocks were ordered from Twist Biosciences and inserted by Gibson assembly (New England Biolabs) into linearized and gel purified (Qiagen) pHR vectors either containing only the plasmid backbone or previously used ectodomains[3]. Plasmid sequences were confirmed by Sanger sequencing over the insert or full plasmid Nanopore sequencing (UC Berkeley Sequencing Facility at Barker Hall).

Gene block sequences are as follows:

| T2A-SV40NLS-GFP | CGGATCAAACAGAAGAAAGCCAAGGGGTCAGAGGGCAGAGGAAGT CTGCTAACATGCGGTGACGTCGAGGAGAATCCTGGCCCACCAAAG AAGAAGCGTAAGGTAATGGTGAGCAAGGGCGAGGAGCTGTTCACC GGGGTGGTGCCCATCCTGGTCGAGCTGGACGGCGACGTAAACGGC CACAAGTTCAGCGTGTCCGGCGAGGGCGAGGGCGATGCCACCTAC GGCAAGCTGACCCTGAAGTTCATCTGCACCACCGGCAAGCTGCCC GTGCCCTGGCCCACCCTCGTGACCACCCTGACCTACGGCGTGCAG TGCTTCAGCCGCTACCCCGACCACATGAAGCAGCACGACTTCTTCA AGTCCGCCATGCCCGAAGGCTACGTCCAGGAGCGCACCATCTTCTT CAAGGACGACGGCAACTACAAGACCCGCGCCGAGGTGAAGTTCGA GGGCGACACCCTGGTGAACCGCATCGAGCTGAAGGGCATCGACTT CAAGGAGGACGGCAACATCCTGGGGCACAAGCTGGAGTACAACTA CAACAGCCACAACGTCTATATCATGGCCGACAAGCAGAAGAACGG CATCAAGGTGAACTTCAAGATCCGCCACAACATCGAGGACGGCAGC GTGCAGCTCGCCGACCACTACCAGCAGAACACCCCCATCGGCGACG GCCCCGTGCTGCTGCCCGACAACCACTACCTGAGCACCCAGTCCGC CCTGAGCAAAGACCCCAACGAGAAGCGCGATCACATGGTCCTGCTG GAGTTCGTGACCGCCGCGGGATCACTCTCGGCATGGACGAGCTGT ACAAGTAAGCCGGCCGCGACTCTAGAGTCGACCTGCAGGCATGCAAG CTTGATATCAAGCTTATCGA |
| --- | --- |

## Lentiviral preparation and cell line generation

Lentivirus was produced by transfecting HEK293T cells with the plasmid of interest, pCMV-dR8.91, and pMD2.G (1.5 μg, 1.33 μg, and 0.167 μg per 35 mm well) using Mirus TransIT-293 Transfection Reagent per manufacturer's protocol. After 60-72 hours, supernatant containing viral particles was harvested and filtered with a 0.45 μm syringe filter (Corning). Supernatant was immediately used for transduction or aliquoted and stored at −80 °C. Cells were seeded at 20% confluency in 35 mm dishes and 0.1-1 mL of filtered viral supernatant was added to the cells. Media containing virus was replaced with fresh growth medium 24 hr post-infection. For suspension cells, cells were resuspended in media containing viral supernatant and 8 μg/mL polybrene (Sigma) and centrifuged at 1000 $g$ and 33 °C for 2 h. Infected cells were imaged to assess transduction efficiency and then used in flow cytometry assays as described above.

## PEG-mediated cell-cell fusion assay

Cells were counted and mixed at 500,000 cells per condition. Each mixture was then centrifuged and washed 2x in serum-free media at 500 $g$ for 5 minutes. Cells were then pelleted, and 500 μL of PEG3.5k (80% w/v in PBS) was added to the pellet and incubated at room temperature for 2 minutes. 10 mL of serum-free media was then added to the mixture, and cells were allowed to recover at room temperature for 30 minutes. The pellet was then washed repeatedly with serum-free media until the pellet became loosened. The pellet was then carefully resuspended in 1 mL of serum-free media and transferred to a 96-well plate to quantify fluorescence in the Attune CytPix flow cytometer.

## p14 cell-cell fusion experiments

**Sample preparation and microscopy.** CellVis 8-well TC-treated chambers were first fibronectin coated by incubating each well with fibronectin diluted 1:50 in PBS for 30 minutes at 37 °C. HEK293T cells were plated one day before transfection and seeded at 20,000 cells/well. The following day, they were transfected with Mirus TransIT 293 according to manufacturer protocol. 19 h post-transfection, cells were incubated with Hoescht and CellMask for 10 minutes at 37 °C, and then transferred to the confocal microscope for imaging using a Nikon W1 spinning disk microscope on an Ti2 body through a 20X air objective (E Plan, NA = 0.40) with a Orca Fusion BT CMOS camera running Nikon Elements software.

**Image analysis.** To analyze cell-cell fusion efficiency, at least 1500 nuclei per well were segmented using the CellPose "nuclei" model based on Hoechst labeling. The CellMask channel was used to identify

| T2A-SV40NLS-miRFP | CGGATCAAACAGAAGAAAGCCAAGGGGTCAGAGGGCAGAGGAAGT CTGCTAACATGCGGTGACGTCGAGGAGAATCCTGGCCCACCAAAG GCATTCGGGACCGCCTCTCATTCGAATTGCGAACATGAAGAGATCC ACCTCGCCGGCTCGATCCAGCCGCATGGCGCGCTTCTGGTCGTCAG CGAACATGATCATCGCGTCATCCAGGCCAGCGCCAACGCCGCGGAA TTTCTGAATCTCGGAAGCGTACTCGGCGTTCCGCTCGCCGAGATCG ACGGCGATCTGTTGATCAAGATCCTGCCCGCATCTCGATCCCACCGC CGAAGGCATGCCGGTCGCGGTGCGCTGCCGGATCGGCAATCCCTCT ACGGAGTACTGCGGTCTGATGCATCGGCCTCCGGAAGGCGGGCTG ATCATCGAACTCGAACGTGCCGGCCCCGTCGATCGATCTGTCAGGCA CGCTGGCGCCGGCCGCTGGAGCGGATCCGCACGGCGGGTTCACTGC GCGCGCTGTGCGATGACACCGTGCTGCTGTTTCAGCAGTGCACCGG CTACGACCGGGTGATGGTGTATCGTTTCGATGAGCAAGGCCACGGC CTGGTATTCTCCGAGTGCCATGTGCCTGGGCTCGAATCCTATTTCG GCAACCGCTATCCGTCGTCGACTGTCCCGCAGATGGCGCGGCAGC TGTACGTGCCGGCAGCGCGTCCGCGTGCTGGTCGACGTCACCTATC AGCCGGTGCCGCTGGAGCCGGGCTGTCGCCGCTGACCGGGCGCG ATCTCGACATGTCGGGCTGCTTCCTGCGCTCGATGTCGCCGTGCCA TCTGCAGTTCCTGAAAGGACATGGGCGTGCGCGCCACCCTGGCGGTG TCGCTGGTGGTCGGCGGCAAGCTGTGGGGCCTGGTTGTCTGTCACC ATTATCTGCCGCGCTTCATCCGTTTCGAGCTGCGGGGCGATCTGCAAA CGGCTCGCCGAAAGGATCGCGACGCGGATCACCGCGCTTGAGAGCT AAGCGGCCGCGACTCTAGAGTCGACCTGCAGGCATGCAAGCTTGAT ATCAAGCTTATCGA |
| --- | --- |

syncytia, either manually for counting experiments or by custom automated MATLAB pipeline for quantifying nuclear expression levels. Each condition was averaged over at least four repeats to determine fusion percentage.

## Crowding sensor measurements

The crowding sensor oligo, 5′-FITC-TTTTTT-biotin-TTT-cholesterol-3′, was ordered from IDT and resuspended at 100 μM. Cells (dissociated with Versene, if necessary) or beads coated with SLBs (assembled as described above and containing a lipid composition of 79% molar fraction DOPC, 1% DOPS, 20% cholesterol from Avanti Polar Lipids) were pelleted and resuspended in PBS. They were then chilled on ice for 15 minutes and incubated with oligo at a concentration of 50 nM. Cells were then washed three times by centrifugation and resuspension and incubated with the desired concentration of Alexa Fluor 647-conjugated anti-biotin antibody BK-1/39 (Santa Cruz Biotechnology, sc-53179) at a cell density of -10000-20000 per 200 μL. Cells were then assayed in a 96-well format using the autosampler of an Attune Cytpix Flow Cytometer with the BRVY configuration and the factory default filters. Data was analyzed using a custom script in MATLAB. The occupancy ratio $\theta$ was calculated by taking the ratio of 647 fluorescence (RL1-H) to 488 fluorescence (BL1-H). The occupancy ratio was then fit to a Langmuir isotherm to calculate the $K_d$, which was normalized to SLB-bead condition.

## Statistics and reproducibility

For all statistically relevant experiments, at least 3 biologically independent replicates (when possible) were performed. No statistical method was used to predetermine sample size. The experiments were not randomized. The investigators were not blinded to allocation during experiments and outcome assessment.

## Reporting summary

Further information on research design is available in the Nature Portfolio Reporting Summary linked to this article.

## Data availability

All data generated in this study have been deposited at the Center for Open Science, under [https://doi.org/10.17605/OSF.IO/AF2Q7], and any further requests may be directed to the corresponding author.

## Code availability

All code associated with this manuscript are available at the Center for Open Science, at [https://doi.org/10.17605/OSF.IO/AF2Q7], and any further requests may be directed to the corresponding author.

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

## Acknowledgements

We would like to thank Pengpeng Bi (University of Georgia) for the myomaker knockout cell line and useful discussions. We would also like to thank Mary West of the QB3 Cell and Tissue Analysis Facility for flow cytometry assistance, and the members of the Fletcher lab for useful discussions. D.A.F. is supported by NIH R01 GM134137, the NSF Center for Cellular Construction (DBI-1548297), and the Miller Institute for Basic Research. D.A.F. is a Chan Zuckerberg Biohub Investigator.

## Author contributions

D.S.W.L., L.F.O., S.S., and D.A.F. designed the research, performed experiments and formal analysis, and wrote the paper.

## Competing interests

The authors declare no competing interest.
