## [Transparent Peer Review file · Nature Communications]

Cell surface crowding is a tunable energetic barrier to cell-cell fusion

Corresponding Author: Dr Daniel Fletcher

Version 0:

Reviewer comments:

Reviewer #1

(Remarks to the Author)

This study uses diverse approaches and experimental models to explore the role of cell surface crowding in cell-cell fusion. The authors demonstrate that at a high enough density, the steric barrier of the proteins at the surface of the membranes suppresses their fusion and suggest that crowding controls fusion efficiency in natural cell-cell fusion processes. The conclusion that raising the steric barrier by boosting protein crowding inhibits fusion processes is consistent with earlier data showing inhibition of liposome fusion by steric effect of pegylated lipids preventing the close contact of liposomes (Tomatsu et al., *J. Mater. Chem.*, 2011,21, 18927) and with inhibition of cell-cell fusion by blocking the displacement of surface glycoproteins from the future fusion site with crosslinking lectins (PMID: 16155572). The novel hypothesis that cells can promote fusion by lowering the cell surface crowding is supported only by experiments with mild trypsin treatment on PEG-induced cell fusion, and, in my opinion, needs additional support. Below I discuss potential limitations of the approaches used in this study.

Specific comments.

1) I have several questions on correlating the results of fusion and crowding assays.

1a. Does the crowding sensor measure the energy to clear from proteins the future fusion site? Taking, as the authors, the diameter of the fusion interface as 0.1 μm , its area of $\sim 0.01 \mu\text{m}^2$ is a very small fraction of the $\sim 1,200 \mu\text{m}^2$ area of the surface of a $\sim 10 \mu\text{m}$ radius cell. Opening this protein depleted patch increases the density of the protein coat at the cell surface by only a 0.001% and this corresponds to binding of just 100 molecules of IgG antibody molecules (each of $\sim 10^{-4} \mu\text{m}^2$ area). It is important to verify that antibody binding in the conditions of the Langmuir isotherm analysis used here to estimate the crowding energy does not itself increase the crowding leading to overestimation of the crowding energy?

1b. In contrast to cell-fusion assays carried out at 37C (p14 fusion) or at the room temperature (PEG induced fusion), crowding sensor measurements were carried out on ice to prevent internalization of the macromolecules. Does the protein crowding depend on the temperature? Can much faster diffusion and ability of the cells to adjust the surface area by endocytosis, release of extracellular vesicles and generation of plasma membrane protrusions at higher temperatures lower the effects of the crowding? Does the crowding density vary between different regions of the same cell under condition of cell fusion experiments (for p14 mediated fusion: attached cells, physiological temperature)?

1c. Fig. 2H,I presents the data of an experiment supporting the hypothesis that lowering the "normal" cell surface crowding promotes fusion (Fig. 2H,I). The evidence for the correlation does not look overwhelming. For 0.05% trypsin, fusion rate is much lower after 10 min treatment than after 5 min treatment. For 0.01% trypsin, what appears to be a major change in ΔU s (10 min vs. 20 min) is not accompanied by a change in fusion. Perhaps the suggested correlation can be strengthened by additional experiments.

1d. A decrease in the crowding during myogenic differentiation of C2C12 cells is an interesting finding consistent with the hypothesis that lower crowding facilitates fusion. However, it is difficult to specifically connect this finding to the development of the fusion competence considering that proliferating and differentiating cells drastically differ in many properties, including, based on proteomic studies, different levels of expression of about 100 proteins.

2) How does the crowding influence fusion?

2a. Does protein crowding influence generation of plasma membrane protrusions, suggested by many studies to play critical

role in cell fusion? Does crowding suppress cell-cell adhesion, i.e. a stage upstream of the fusion pathway?

2b. Fig. 3C. In Fig. 3E, the efficiency of transfection with nuclear GFP is much lower for the construct with SIRP α than for the construct with Fib1L. Does expression of SIRP α bystanders in Fig. 3C decrease cell surface expression of p14?

3) Fig. 1B What do we see in the image on the right? Is it already after tearing the beads from SLB or before? How was the efficiency of tearing the beads from the SLB evaluated? Have you verified the lack of the red spots on SLB after tearing away the beads? Can the separating beads peel off the lipids from SLB as vesicles? I do see a red bead with green spot. Will flow analysis show it as fused?

4) Fig. 2C. Why are the bilayers in SLB and cell cartoon shown as striped (pink then green than pink again)? Pink in the inner monolayer of SLB in this cartoon suggests complete fusion. Can you distinguish hemifusion from fusion in the presented experiments?

5) How exactly were the fusion rates scored for HEK cells and for Jurkat cells? Figure legend states "... and analyzed by microscopy or flow cytometry" (line 305). Which one was used in the data presented? How were the fused cells (syncytia containing both H2B-miRFP670 and H2B-GFP labeled nuclei) were distinguished from aggregates of differently labeled cells if it was done by flow cytometry?

Minor comments

1) "... containing 20% DOPE (1,2-Dioleoyl-sn-98 glycerol-3-phosphoethanolamine), a lipid that induces positive curvature. " The authors probably meant negative curvature. (According to a common convention, curvature generated by the lipids with relatively small headgroup and bulky hydrophobic tails, such as PE, is referred to as negative.)

2) In Supplemental Info, a paragraph describing PEG-mediated cell-cell fusion assay is duplicated.

3) Something seems to be missing in the sentence "Similarly, we showed that the cell surface of developing myoblasts is reduced by 1kBT following differentiation."

(Remarks on code availability)

Reviewer #2

(Remarks to the Author)

This manuscript investigates whether cell surface crowding impacts cell-cell fusion, which is a critical process for many developmental and regenerative systems. The premise is that close membrane apposition is a barrier to cell fusion that must be overcome and part of that process may require clearing of proteins in the bilayer. Currently there is minimal experimental evidence in support of the membrane crowding hypothesis. Using multiple membrane fusion systems induced by PEG, developmental fusogens, in vitro reconstitution to demonstrate the cell surface crowding is a barrier to fusion. This work is highly original, the data are solid, the paper is well-written and the claims are consistent with the data. Below are some relatively minor comments the authors could consider to make their manuscript more accessible to readers. Deeper explanations for the experimental conditions should give readers more insight into the rationale and interpretations.

1. Fig. 1 shows that fibronectin domains reduce in vitro fusion in a density dependent manner, but only when longer domains are used (3 and 5). The data support the idea that density and size of surface proteins is important for regulation of fusion. Is it possible for authors to give some context about how the densities and sizes used in this in vitro assay can be related to cellular systems?

2. The transition to the crowding sensor (Fig. 2F) is abrupt. So that readers from broad backgrounds can understand the manuscript it would be great if the authors could explain this sensor and show quality control data for its use. I realize the sensor is mentioned in the introduction, but it might be good to add a sentence explaining its use in the results. I think there could be questions if the sensor integrates at different concentrations between cells or integrates within different domains of the membrane, which could impact the cell surface crowding calculations. It is also not clear why the authors chose to use K562 cells since their crowding does not seem dramatically different than crowding in Jurkat cells. Does the trypsin treatment effectively reduce cell surface crowding? I think it may be possible to use their sensor to definitively establish that their manipulation (trypsin) is impacting the intended cellular effect (reduction of cell surface proteins).

3. Does SIRP α have an effect on cells other than increasing surface crowding? It is possible that this is in their previous papers and if so it may be sufficient to add a sentence to the text.

4. I am not sure readers will understand the quantification of biased fusion in Fig. 3. The quantification in F does not seem to represent the effect observed in E. I think there is an underestimation of the effect because the authors do not know how many nuclei within the syncytia correspond to the GFP+ population.

5. For myoblast cell surface crowding, it is unclear which myogenic state the authors are comparing. Flow cytometry is designed to capture cells based on size and myotubes are typically excluded, so at day 6 they may be assaying C2C12 reserve cells (mononucleated cells that do not differentiate). It may be more appropriate to do quantification/statistics between day 0 and day 2 when there are more mononucleated differentiated cells that have not transitioned to myotubes. Also, when is the cell size sensor added to the cultures for each timepoint? I think it will be important for readers to know these kinetics to better interpret the results.

(Remarks on code availability)

n/a

Version 1:

Reviewer comments:

Reviewer #1

(Remarks to the Author)

The authors mostly addressed my comments/questions. The revised paper will be an important contribution to understanding the mechanisms controlling membrane fusion. My only remaining suggestion is related to the Authors Response to my 1st round comment 1b. "In contrast to cell-fusion assays carried out at 37C (p14 fusion) or at the room temperature (PEG induced fusion), crowding sensor measurements were carried out on ice to prevent internalization of the macromolecules."

Yes, in units of absolute temperature, the change from 4C to 37C is rather small but we are discussing the properties of membranes of living cells. As the authors note, it is difficult to account for any temperature dependence of biological processes. Many papers report considerable and, in some cases, puzzling (for instance, PMID: 4833254) changes in protein mobility and many other properties of plasma membrane in this temperature range. This study presents correlations, and even, impressive quantitative correlations, between changes in different membrane properties: crowding and propensity to fuse. In my view, the fact that these properties are evaluated at different temperatures (4C vs 22C or 37C) is an important potential limitation (perhaps currently unavoidable) and this should be explicitly noted. The added comment in the discussion on the yet unexplored biological mechanisms controlling the crowding helps but does not inform the readers that the presented analysis in effect assumes that the plasma membrane crowding does not change in this temperature range.

(Remarks on code availability)

Reviewer #2

(Remarks to the Author)

I don't have any further concerns or suggestions.

(Remarks on code availability)

Response to Reviewers

“Cell surface crowding is a tunable energetic barrier to cell-cell fusion”

Daniel S.W. Lee*, Liya F. Oster*, Sungmin Son, and Daniel A. Fletcher

We thank the reviewers for their careful reading, constructive critique, and generous assessment of our manuscript. Below, we address their comments point-by-point, with our responses in blue text and the reviewer comments in black.

Reviewer #1 (Remarks to the Author):

This study uses diverse approaches and experimental models to explore the role of cell surface crowding in cell-cell fusion. The authors demonstrate that at a high enough density, the steric barrier of the proteins at the surface of the membranes suppresses their fusion and suggest that crowding controls fusion efficiency in natural cell-cell fusion processes. The conclusion that raising the steric barrier by boosting protein crowding inhibits fusion processes is consistent with earlier data showing inhibition of liposome fusion by steric effect of pegylated lipids preventing the close contact of liposomes (Tomatsu et al., *J. Mater. Chem.*, 2011,21, 18927) and with inhibition of cell-cell fusion by blocking the displacement of surface glycoproteins from the future fusion site with crosslinking lectins (PMID: 16155572). The novel hypothesis that cells can promote fusion by lowering the cell surface crowding is supported only by experiments with mild trypsin treatment on PEG-induced cell fusion, and, in my opinion, needs additional support. Below I discuss potential limitations of the approaches used in this study.

We appreciate the reviewer’s comments and have included citations to the above-mentioned publications in our updated manuscript.

Specific comments.

1) I have several questions on correlating the results of fusion and crowding assays.

1a. Does the crowding sensor measure the energy to clear from proteins the future fusion site? Taking, as the authors, the diameter of the fusion interface as 0.1 μm , its area of $\sim 0.01 \mu\text{m}^2$ is a very small fraction of the $\sim 1,200 \mu\text{m}^2$ area of the surface of a $\sim 10 \mu\text{m}$ radius cell. Opening this protein depleted patch increases the density of the protein coat at the cell surface by only a 0.001% and this corresponds to binding of just 100 molecules of IgG antibody molecules (each of $\sim 10^{-4} \mu\text{m}^2$ area). It is important to verify that antibody binding in the conditions of the Langmuir isotherm analysis used here to estimate the crowding energy does not itself increase the crowding leading to overestimation of the crowding energy?

The crowding sensor, as previously described (Takatori et al. 2023), measures the average energy required for a single IgG protein to access the cell surface by calculating the change in binding energy (ΔU_s) between the sensor on a cell surface on a bare surface.

As the reviewer points out, the estimated size of the fusion patch is approximately 100 times the size of an IgG protein. We estimate the energetic cost of clearing the fusion patch by linearly scaling the energetic cost of the IgG patch by area, making the assumption that clearing a 100nm versus a 10nm diameter area does not cause compression or other effects on the remaining cell surface. We expect such perturbations to be small since, as the reviewer also identifies, this is a small portion of the total area of the cell.

Along the same lines, it is important to verify that the addition of the IgG protein during our measurements does not itself increase crowding and likewise does not violate assumptions of the Langmuir isotherm, since at high sensor incorporation and high antibody concentrations there may be enough antibodies to affect surface crowding. However, based upon a fluorescence calibration using reference beads (AccuCheck ERF Reference Particles, Thermo Fisher), we estimate an average of ~ 100 oligos/ μm^2 , compared to an average of $\sim 20,000$ native proteins per square micron. Even if the sensors were entirely saturated by antibodies (though for most relevant antibody concentrations, we estimate a much lower bound fraction, closer to $\sim 10\%$), this would result in an area fraction change of about 1%, suggesting that the contribution of our sensor be minimal even when bound.

Furthermore, if this were the case, we would observe non-linearities when plotting sensor intensity against antibody binding. In particular, there would be proportionally less antibody binding at higher sensor incorporation levels.

Ordinarily, we expect $\frac{[I_{ab}]}{[I_{oligo}]} \sim \frac{c}{c+K_d}$, and since c and K_d are fixed, we simply observe a line with constant slope, i.e., $I_{ab} \sim I_{oligo}$, consistent with our data (e.g., Fig S3A, Fig R1). If the sensor introduced additional crowding, increasing the oligo intensity would result in a higher K_d , and thus $K_d \sim I_{oligo}$, giving $I_{ab} \sim \frac{cI_{oligo}}{c+K_d^0 I_{oligo}}$, i.e., a sigmoid.

To further validate that making the crowding measurement does not introduce additional crowding, we plotted antibody versus oligo concentration from example replicates with wild-type K562 cells at $0.05\mu\text{g/mL}$ and $0.0125\mu\text{g/mL}$, respectively, and fit each concentration to a linear and sigmoidal (see Fig. R1; black lines indicate linear fit with a fixed zero intercept, and red lines indicate a sigmoidal fit as described above). While the linear model appears to be a better fit visually, we also compared them using an F test to correct for overfitting. In each instance, we calculated a p value $\ll 0.05$ against the null hypothesis that a sigmoidal fit is

significantly better than a linear one, i.e., the linear model is a better fit with high statistical certainty after correcting for overfitting.

Fig. R1: Example flow cytometry data for the crowding sensor used in wild-type K562 cells, comparing incorporation level (488 intensity) to antibody binding (647 intensity), fitting to both a sigmoidal (red dashed line) and a linear (black dashed line) model, for **A**, 0.05 μ g/mL and **B**, 0.0125 μ g/mL. In both instances, we find that the black line fits the data, suggesting that the sensor is not inducing crowding and the assumptions of the Langmuir isotherm model hold, since we observe a constant sensor occupancy (i.e., one slope) for each antibody concentration condition.

1b. In contrast to cell-fusion assays carried out at 37C (p14 fusion) or at the room temperature (PEG induced fusion), crowding sensor measurements were carried out on ice to prevent internalization of the macromolecules. Does the protein crowding depend on the temperature? Can much faster diffusion and ability of the cells to adjust the surface area by endocytosis, release of extracellular vesicles and generation of plasma membrane protrusions at higher temperatures lower the effects of the crowding? Does the crowding density vary between different regions of the same cell under condition of cell fusion experiments (for p14 mediated fusion: attached cells, physiological temperature)?

We do not anticipate that changes in temperature affect crowding substantially. Considering the crowding energy, $\Delta U_s = k_B T \log \left(\frac{K_d}{K_d^{bare}} \right)$, temperature appears as a linear scaling term. Similarly, the timescale of protein mixing by diffusion on the cell surface is generally on the timescale of seconds, and it is governed by the diffusion coefficient, whose dependence on temperature is $D \sim \frac{1}{k_B T}$. In each of these cases, T is in units of absolute temperature. Given that room temperature corresponds to about 293K, 37C is about 310K, and 4C is 277K, being at room temperature increases the value of both the diffusion coefficient and the crowding energy by

about 7% compared to 4C, while being at 37C increases the value of the diffusion coefficient by 12%; both of these are relatively small effects considering the large changes in crowding between conditions (i.e., on the order of a $1k_bT$ change out of $\sim 2-3k_bT$ in ΔU_s due to trypsinization, myoblast differentiation, or SIRPA ectodomain overexpression).

However, it is more difficult to account for any temperature dependence of biological processes that might regulate the crowding of the cell surface, and this is the subject of ongoing investigations in the lab. We have added a sentence in the text addressing these possibilities.

Finally, our investigations using the crowding sensor have not identified major heterogeneities in crowding on the cell surface, but this is also a subject of ongoing investigation.

1c. Fig. 2H,I presents the data of an experiment supporting the hypothesis that lowering the “normal” cell surface crowding promotes fusion (Fig. 2H,I). The evidence for the correlation does not look overwhelming. For 0.05% trypsin, fusion rate is much lower after 10 min treatment than after 5 min treatment. For 0.01% trypsin, what appears to be a major change in ΔU_s (10 min vs. 20 min) is not accompanied by a change in fusion. Perhaps the suggested correlation can be strengthened by additional experiments.

We thank the reviewer for this suggestion. We have now repeated this experiment 3-4 times per condition and per assay and found that the effect of trypsinization was relatively consistent across timepoints for each dosage of trypsin, which is consistent with the observation that the fusion probabilities also are also clustered by dosage rather than time. We have plotted a new version of Fig2G/H with the additional replicates. We note that the Pearson’s correlation in the plot shown is approximately $R=-0.76$, suggesting a strong degree of negative correlation, and that an F test against the null hypothesis that a constant model (i.e., no correlation) can be rejected with $p=0.04$.

1d. A decrease in the crowding during myogenic differentiation of C2C12 cells is an interesting finding consistent with the hypothesis that lower crowding facilitates fusion. However, it is difficult to specifically connect this finding to the development of the fusion competence considering that proliferating and differentiating cells drastically differ in many properties, including, based on proteomic studies, different levels of expression of about 100 proteins.

We thank the reviewer for this insight. In the context of our assay, we are evaluating only the change in crowding, an emergent property that is itself a product of the numerous changes that occur within the cell during differentiation. The data we show demonstrate a correlation consistent with our hypothesis, indicating that these myriad changes—among other important changes in protein expression and signaling—together lead to a coordinated change in crowding that we argue helps to facilitate fusion.

Furthermore, the data we show examining the fusion-incompetent myomaker knockdown cell line support this hypothesis that myoblast differentiation drives these crowding changes, independent of myomaker or the fusion event itself. We have added a paragraph addressing this in the discussion.

Which gene expression or developmental changes lead to lower crowding is a complex and very interesting question, particularly in this context. As part of a separate project, we are in the process of investigating what genes and biological processes contribute to crowding via genome-wide CRISPR inhibition assays.

2) How does the crowding influence fusion?

2a. Does protein crowding influence generation of plasma membrane protrusions, suggested by many studies to play critical role in cell fusion? Does crowding suppress cell-cell adhesion, i.e. a stage upstream of the fusion pathway?

While we and others have shown that local crowding can drive membrane curvature (Schmid et al. 2016; Shurer et al. 2019), we did not observe increased protrusions associated with higher surface crowding in the cell types we explored in this manuscript. It is certainly possible that we are not examining the cell surface with sufficient resolution to resolve protrusions like those observed in the context of cell-cell fusion. However, if increased crowding increases plasma membrane protrusions, they are not sufficient to increase cell-cell fusion, as we find decreased fusion with increased surface crowding.

We do expect that increased crowding could suppress cell-cell adhesion, especially for short or membrane-proximal receptors. This could be an important factor in reduced fusion with crowding, in addition to the increased energetic penalty associated with bringing the two more crowded membranes into close contact. However, we note that in the context of our PEG-based fusion assay, the cells are pelleted before adding the PEG, so they are already in relatively close contact. In addition, no specific adhesion is known to be required for p14-mediated cell-cell fusion.

2b. Fig. 3C. In Fig. 3E, the efficiency of transfection with nuclear GFP is much lower for the construct with SIRP α than for the construct with Fib1L. Does expression of SIRP α bystanders in Fig. 3C decrease cell surface expression of p14?

We thank the reviewer for bringing up this important point about p14 expression. To address this question, we have performed an additional control experiment co-transfecting p14-GFP with SIRP α ectodomain (ecSIRP α) or Fib1L bystanders. While the p14-GFP construct is fusion

competent, it is less fusogenic than the unlabeled wild-type (Chan et al. 2020), which is why we utilized the wild-type p14 in the experiments in the manuscript. However, the p14-GFP allows us to quantify the expression level of p14 before fusion in the context of co-expressed proteins. Following equimolar co-transfection with p14-GFP and a bystander protein and incubation for 22 hours, HEK cells were dissociated with EDTA and assayed for GFP expression by flow cytometry.

The results are shown in Fig R2, in which we observe that the p14 transfection efficiency is higher for ecSIRP α cotransfection and slightly lower for Fib1L cotransfection compared to the wild type control, while the expression levels in the transfected cells were nearly identical across the three conditions. The differences between conditions for either cotransfection efficiency or expression level were not statistically significant by one-way ANOVA test followed by a Tukey's test for multiple comparisons.

Fig R2: Cells were co-transfected with p14-GFP and Fib1L or ecSIRP α constructs. They were dissociated with EDTA 22 hours following transfection and assayed by flow cytometry to calculate **A**, the percentage of GFP positive cells and **B**, the average expression level of GFP positive cells over three biologically independent replicates. Error bars represent standard deviation over three biologically independent replicates. These quantities were compared across the three conditions by one-way ANOVA test followed by a Tukey's test for multiple comparisons.

3) Fig. 1B What do we see in the image on the right? Is it already after tearing the beads from SLB or before? How was the efficiency of tearing the beads from the SLB evaluated? Have you verified the lack of the red spots on SLB after tearing away the beads? Can the separating beads peel off the lipids from SLB as vesicles? I do see a red bead with green spot. Will flow analysis show it as fused?

We apologize for the ambiguity in the description of the assay in Fig. 1B. In this assay, we evaluate the lipid mixing between the SLB on the bead (labeled red, with Rhodamine-PE) and the planar SLB (labeled green, with Atto488). On the right, we include an image of the beads post-centrifugation but before being separated from the planar SLB. To verify that the beads were removed from the SLB, we imaged it following removing the beads and confirmed that less than 1% remain. We observe no red spots on the SLB; however, there may be green SUVs that remain after the formation of the SLB (despite washing) or degradation of the SLB over time that releases membrane fragments, and those are sometimes observed stuck to the red beads. An example of that is seen on the right of Fig 2C, as the reviewer noted. However, the presence of small vesicles associated with the red fluorescence would not be counted as fused in the flow analysis because we use a threshold of 50% gain of the green signal (relative to the maximum green signal) to consider a bead fused.

We have updated the schematic in Figure 1B to clarify the mixing of membranes. We have also added clarifying text to the methods regarding bead removal from the SLB.

4) Fig. 2C. Why are the bilayers in SLB and cell cartoon shown as striped (pink then green than pink again)? Pink in the inner monolayer of SLB in this cartoon suggests complete fusion. Can you distinguish hemifusion from fusion in the presented experiments?

We agree that the stripes are confusing (in both Fig 2C and Fig 1B). We have updated both figures. Fig. 2C now shows a fully fused state in which the cell membrane has mixed with the supported bilayer, resulting in a loss of red fluorescence and gain of green fluorescence.

Hemifusion and full fusion could potentially be distinguished in the bead and cell experiments from the amount of fluorescence gain and loss. For example, if only the outer leaflet of the bead experiment were fused with the outer leaflet of the SLB, then we would expect to see beads that have lost half of their red fluorescence and gained half of maximum green fluorescence. However, we do not see a significant population of hemifused beads in our flow assay, potentially because of the transience of the hemifusion state and complications of lipid-glass interaction of the SLB.

5) How exactly were the fusion rates scored for HEK cells and for Jurkat cells? Figure legend states "... and analyzed by microscopy or flow cytometry" (line 305). Which one was used in the data presented? How were the fused cells (syncytia containing both H2B-miRFP670 and H2B-GFP labeled nuclei) were distinguished from aggregates of differently labeled cells if it was done by flow cytometry?

Fusion rates were scored only for K562 cells—the Jurkat example was used in an illustrative fashion to corroborate that our assay does achieve full mixing. Following hybridoma formation, cells were heavily pipetted and vortexed to break up any aggregates. Fused cells were identified

using imaging flow cytometry (Fig S2). We utilized an automated size threshold to identify fused cells, and then we manually inspected each image to verify that the cells were fully fused as opposed to merely adhered. Importantly, the K562 cells do not clump or aggregate in suspension, making it a useful cell for this assay. We have clarified this point in the text.

Minor comments

1) "... containing 20% DOPE (1,2-Dioleoyl-sn-98 glycerol-3-phosphoethanolamine), a lipid that induces positive curvature. " The authors probably meant negative curvature. (According to a common convention, curvature generated by the lipids with relatively small headgroup and bulky hydrophobic tails, such as PE, is referred to as negative.)

We thank the reviewer for their detailed reading of the manuscript and have corrected this.

2) In Supplemental Info, a paragraph describing PEG-mediated cell-cell fusion assay is duplicated.

We thank the reviewer for catching this mistake and have removed the repeated paragraph.

3) Something seems to be missing in the sentence "Similarly, we showed that the cell surface of developing myoblasts is reduced by $1kBT$ following differentiation."

We have edited the text to remedy this issue.

Reviewer #2 (Remarks to the Author):

This manuscript investigates whether cell surface crowding impacts cell-cell fusion, which is a critical process for many developmental and regenerative systems. The premise is that close membrane apposition is a barrier to cell fusion that must be overcome and part of that process may require clearing of proteins in the bilayer. Currently there is minimal experimental evidence in support of the membrane crowding hypothesis. Using multiple membrane fusion systems induced by PEG, developmental fusogens, in vitro reconstitution to demonstrate the cell surface crowding is a barrier to fusion. This work is highly original, the data are solid, the paper is well-written and the claims are consistent with the data. Below are some relatively minor comments the authors could consider to make their manuscript more accessible to readers. Deeper explanations for the experimental conditions should give readers more insight into the rationale and interpretations.

We thank the reviewer for their generous assessment of our work and careful reading of the manuscript. We address their comments below.

1. Fig. 1 shows that fibronectin domains reduce in vitro fusion in a density dependent manner, but only when longer domains are used (3 and 5). The data support the idea that density and size of surface proteins is important for regulation of fusion. Is it possible for authors to give some context about how the densities and sizes used in this in vitro assay can be related to cellular systems?

We thank the reviewer for their suggestion. The fibronectin domains used in these in vitro assays are chosen to be representative of surface proteins expressed in cells; typical cell surfaces have a density of ~20000 proteins per square micron (Itzhak et al. 2016). The densities we reach in our in vitro assays (~10000 per square micron) tend to be lower than those typical of cells, due to limitations on the nickel lipid content of the SLBs and instability of the SLBs at very high labeling densities. The average protein height on the cell surface is ~12nm, while the height of the Fib5L domain is ~10nm (20nm fully extended) (Son et al. 2020), suggesting that the sizes of the proteins used are similar to those in the cell. We have updated the main text with this useful context.

2. The transition to the crowding sensor (Fig. 2F) is abrupt. So that readers from broad backgrounds can understand the manuscript it would be great if the authors could explain this sensor and show quality control data for its use. I realize the sensor is mentioned in the introduction, but it might be good to add a sentence explaining its use in the results. I think there could be questions if the sensor integrates at different concentrations between cells or integrates within different domains of the membrane, which could impact the cell surface crowding calculations. It is also not clear why the authors chose to use K562 cells since their crowding

does not seem dramatically different than crowding in Jurkat cells. Does the trypsin treatment effectively reduce cell surface crowding? I think it may be possible to use their sensor to definitively establish that their manipulation (trypsin) is impacting the intended cellular effect (reduction of cell surface proteins).

We thank the reviewer for this helpful comment. We have moved the panel comparing crowding across suspension cell types to Fig S3 for clarity, and we edited the text to reflect that we selected K562s for their high crowding, large size, and propensity to not aggregate compared to Jurkats. Trypsin treatment is indeed meant to reduce cell surface crowding in this context. We show in Fig. 2G that trypsin lowers the crowding (expressed as changes in ΔU_s), which is also used in the x axis of the plot in Fig 2H.

3. Does SIRP α have an effect on cells other than increasing surface crowding? It is possible that this is in their previous papers and if so it may be sufficient to add a sentence to the text.

We do not observe any differences between the cells overexpressing the SIRP α ectodomain and wild type cells in terms of morphology or health. This lack of impact on the cell is likely because the construct we used only expresses the transmembrane and ectodomain regions of SIRP α (residues 25-389), without the cytoplasmic domain (residues 395-504) that contain the signaling ITIM domain and associated SH2- and SH3- binding motifs. We have clarified this by replacing mentions of this construct with “ecSIRP α ”, rather than simply “SIRP α ” in the manuscript, and we have added a sentence in the text addressing this point.

4. I am not sure readers will understand the quantification of biased fusion in Fig. 3. The quantification in F does not seem to represent the effect observed in E. I think there is an underestimation of the effect because the authors do not know how many nuclei within the syncytia correspond to the GFP+ population.

Because the NLS-GFP signal diffuses within the syncytia following fusion, we are unable to directly quantify the number of nuclei expressing it and therefore correct for expression level by normalizing by average fluorescent signal. We have used a conservative estimate of the fusion bias and agree that this may underestimate the effect. We have added text to clarify this point.

5. For myoblast cell surface crowding, it is unclear which myogenic state the authors are comparing. Flow cytometry is designed to capture cells based on size and myotubes are typically excluded, so at day 6 they may be assaying C2C12 reserve cells (mononucleated cells that do not differentiate). It may be more appropriate to do quantification/statistics between day 0 and day 2 when there are more mononucleated differentiated cells that have not transitioned to myotubes. Also, when is the cell size sensor added to the cultures for each timepoint? I think it will be important for readers to know these kinetics to better interpret the results.

We thank the reviewer for the suggestion. In Fig. 4C, we do observe a decrease in crowding between day 0 and day 2 in the wild-type fusion experiment, and that decrease in crowding continues to increase through day 6 for the cells we analyze. The reviewer is correct that some multinucleated cells will not be included in our day 6 analysis due to their size. Our concern, shared by the reviewer, that excluding such cells could lead to an underestimate of the change in crowding due to differentiation led us to carry out the same fusion experiments with myomaker knock-out cells (which will undergo differentiation but not fuse). As shown in Fig 4D, we observe a similar decrease in crowding through day 6, suggesting that the fused cells missing from the analysis in the wild-type experiment would not significantly change our results.

Regarding the timepoint of sensor integration, the cells are differentiated for the duration described in a staggered fashion such that they can be simultaneously assayed by flow, and all cells are incubated with the sensor simultaneously as an endpoint assay. When the assay is performed, all the cells are dissociated with EDTA and incubated with the crowding sensor on ice. The antibody is then added and the cells are analyzed by flow cytometry. We have added text to clarify this.

Reviewer #2 (Remarks on code availability):

n/a

Reviewer Response References

- Bakalar, M. H., A. M. Joffe, E. M. Schmid, S. Son, M. Podolski, and D. A. Fletcher. 2018. "Size-Dependent Segregation Controls Macrophage Phagocytosis of Antibody-Opsonized Targets." *Cell* 174 (1). <https://doi.org/10.1016/j.cell.2018.05.059>.
- Chan, Ka Man Carmen, Sungmin Son, Eva M. Schmid, and Daniel A. Fletcher. 2020. "A Viral Fusogen Hijacks the Actin Cytoskeleton to Drive Cell-Cell Fusion." *eLife* 9 (May). <https://doi.org/10.7554/eLife.51358>.
- Joffe, A. M., M. H. Bakalar, and D. A. Fletcher. 2020. "Macrophage Phagocytosis Assay with Reconstituted Target Particles." *Nature Protocols* 15 (7). <https://doi.org/10.1038/s41596-020-0330-8>.
- Itzhak, Daniel N., Stefka Tyanova, Jürgen Cox, and Georg Hh Börner. 2016. "Global, Quantitative and Dynamic Mapping of Protein Subcellular Localization." *eLife* 5 (June). <https://doi.org/10.7554/eLife.16950>.
- Schmid, E. M., M. H. Bakalar, K. Choudhuri, J. Weichsel, H. Ann, P. L. Geissler, M. L. Dustin, and D. A. Fletcher. 2016. "Size-Dependent Protein Segregation at Membrane Interfaces." *Nature Physics* 12 (7). <https://doi.org/10.1038/nphys3678>.
- Shurer, C. R., J. C. Kuo, L. M. Roberts, J. G. Gandhi, M. J. Colville, T. A. Enoki, H. Pan, et al. 2019. "Physical Principles of Membrane Shape Regulation by the Glycocalyx." *Cell* 177 (7). <https://doi.org/10.1016/j.cell.2019.04.017>.
- Son, Sungmin, Sho C. Takatori, Brian Belardi, Marija Podolski, Matthew H. Bakalar, and Daniel A. Fletcher. 2020. "Molecular Height Measurement by Cell Surface Optical Profilometry (CSOP)." *Proceedings of the National Academy of Sciences* 117 (25): 14209–19.
- Takatori, Sho C., Sungmin Son, Daniel S. W. Lee, and Daniel A. Fletcher. 2023. "Engineered Molecular Sensors for Quantifying Cell Surface Crowding." *Proceedings of the National Academy of Sciences* 120 (21): e2219778120.